# Evaluation of Left Atrial Electromechanical Delay and Left Atrial Phasic Functions in Patients Undergoing Treatment with Cardiotoxic Chemotherapeutic Agents

**DOI:** 10.3390/medicina60091516

**Published:** 2024-09-18

**Authors:** Ömer Kertmen, Murat Akcay

**Affiliations:** 1Department of Cardiology, Amasya University, Sabuncuoğlu Şerefeddin Training and Research Hospital, 05100 Amasya, Turkey; 2Department of Cardiology, Faculty of Medicine, Ondokuz Mayis University, 55270 Samsun, Turkey; drmuratakcay@hotmail.com

**Keywords:** cancer, chemotherapy, cardiotoxicity, left atrial mechanical functions, atrial electromechanical delay

## Abstract

**Background:** The aim of this study is to evaluate atrial involvement by comparing pre- and post-chemotherapy left atrial mechanical and electromechanical parameters in patients treated with cardiotoxic chemotherapeutic agents. **Methods:** We designed our study as a prospective cohort study. Sixty-eight female patients between the ages of 18 and 50, scheduled for treatment with cardiotoxic chemotherapeutic agents, were included in our study. Atrial mechanical functions and electromechanical parameters were examined and compared with basic echocardiographic parameters before and after chemotherapy. **Results:** The mean age of the patients was 41.6 ± 7.9 years. After chemotherapy, lateral PA, septal PA, and tricuspid PA durations showed a significant increase (*p* < 0.001), but there were no statistically significant changes in the left intra-atrial electromechanical delay, the right intra-atrial electromechanical delay, or the interatrial electromechanical delay values. Following treatment, LAVmax, LAVmin, and LApreA significantly increased (*p* < 0.001). Additionally, the left atrial passive and active emptying volumes increased (*p* < 0.001), while the reservoir and pump (active emptying) functions decreased (with *p*-values of 0.03 and 0.01, respectively). The passive emptying function, however, showed no significant change (*p* = 0.65). Decreases in LVEF were observed, while LVEDD, LVESD, IVS, PW, and LA diameters increased (*p*-value of 0.02 for IVS and <0.001 for the others). **Conclusions:** Significant impairment of atrial mechanical functions and electromechanical parameters was observed after treatment with cardiotoxic chemotherapeutic agents. This suggests an elevated likelihood of atrial arrhythmia linked to the use of cardiotoxic chemotherapeutic agents.

## 1. Introduction

The negative effects of chemotherapeutic agents on the left ventricle, and diagnostic and treatment methods for these effects, have been the subject of many studies [1]. 

However, there are not enough publications investigating the effects of these agents on the atrium. Nevertheless, the myocardial damage caused by cardiotoxic chemotherapeutics, both through the formation of free radicals and direct cytotoxic effects, can harm the cardiovascular system as a whole [2].

Although the relationship between chemotherapeutic agents and atrial arrhythmia is hypothetical and needs to be established with larger scale studies, it has been evaluated pathophysiologically by a few important studies [1,2,3,4]. For example, in a study of mice injected with doxorubicin, the left atrial reservoir and conduit function decreased after chemotherapy. And this effect developed long before the ventricular effect developed Doxorubicin-induced af has been linked to structural remodeling (cardiomyocyte death, hypotrophy, and vacuolization) and electrical remodeling (reduction and redistribution of connexin 43) in the atria [5]. Alexandre and colleagues, in a pharmacovigilance study using the World Health Organization individual case safety report database, VigiBase^®^, determined that 19 different chemotherapeutic agents increased the risk of atrial fibrillation. They argued that this effect was due to the inflammatory effects, physiological stress, malnutrition, and free radical damage to the left atrium and sinoatrial node caused by these agents [6].

Atrial fibrillation, being the most common tachyarrhythmia necessitating treatment, presents a notable public health concern due to its relationship with ischemic stroke, thromboembolism, tachycardia-induced cardiomyopathy, reduced functional capacity, frequent hospitalizations, and increased morbidity/mortality [3]. Recent studies have shown that any defect in the atrial conduction system renders patients highly susceptible to the development of atrial fibrillation. Furthermore, studies have shown a marked rise in the likelihood of developing atrial arrhythmia linked to mechanical and electromechanical alterations that coincide with an enlargement of the left atrial volume [4,5].

The combination of conventional echocardiography with Doppler imaging has been observed to provide comparable outcomes when evaluating atrial mechanical functions and electromechanical parameters as compared to gold-standard imaging techniques like cardiac magnetic resonance and cardiac computed tomography. Current data support the safe use of echocardiography in left atrial evaluation as it is cheaper, more repeatable, and more accessible than other imaging modalities [6].

In our study, we aimed to compare pre- and post-treatment echocardiographic atrial mechanical functions and electromechanical parameters in patients scheduled for treatment with cardiotoxic chemotherapeutic agents. With this easily accessible and cheap method we have planned to show a way to help early detection of post-chemotherapeutic atrial arrhythmias.

## 2. Methods

### 2.1. Study Protocol

This study included 68 female participants newly diagnosed with breast cancer or lymphoma who were scheduled for chemotherapeutic treatments by the Department of Medical Oncology at Ondokuz Mayis University. The patients’ age, height, body weight, ongoing medications, cardiovascular risk factors, systolic and diastolic blood pressure, heart rate, physical assessment, fundamental laboratory parameters, and chronic diseases were recorded. The appropriate formulas were applied to estimate body mass index (BMI) and body surface area (BSA). Exclusion criteria were determined as coronary artery and severe heart valve diseases, heart failure, diabetes, hypertension, hyperlipidemia, smoking, and alcohol consumption. The baseline clinical and echocardiographic parameters of all patients were evaluated before chemotherapy was administered. All of these echocardiographic parameters were evaluated again, 1 month after the chemotherapy protocol was completed.

The study received approval from the ethics committee at the Faculty of Medicine, Ondokuz Mayis University (Approval No. 2023/425), and conformed to the principles outlined in the Declaration of Helsinki (2013 edition). All participants were informed about the study’s details, and written consent was obtained.

### 2.2. Standard Echocardiography

All patients underwent an echocardiographic assessment while positioned on their left side and supine, using two-dimensional Doppler, tissue Doppler, and precordial M-mode echocardiography techniques from both parasternal and apical windows. The examinations were carried out by the same operator, utilizing a Vivid 7 device (equipped with a 3.5 MHz phased array transducer; GE Medical System, Horten, Norway). Measurements of the interventricular septum (IVS) thickness, the posterior wall (PW) thickness, the left ventricular end-systolic diameter (LVESD), the left ventricular end-diastolic diameter (LVEDD), and the left atrial anteroposterior diameter (LA) were acquired using the M-mode method. The left ventricular ejection fraction (LVEF) was calculated based on two- or apical four-chamber views utilizing the modified Simpson technique.

For optimal filling velocity data, the pulsed Doppler sampling was taken from the mitral valve leaflets’ tips at the apical four-chamber view. Myocardial velocity patterns of the mitral annuli were obtained. The highest early diastolic velocities (e’), late diastolic velocities (a’), and systolic velocities (s’) were calculated [7]. The obtained data were averaged by considering at least three measurements for each value.

### 2.3. Assessment of Atrial Electromechanical Delay

Tissue Doppler echocardiography was performed using transducer frequencies ranging from 3.5 to 4.0 MHz, adjusting the filters of the spectral pulsed Doppler signal to achieve a Nyquist limit of 15 to 20 cm/s, and applying minimal optimal gain settings. In the apical four-chamber view, the pulsed Doppler sample volume was placed at the septal, lateral mitral annulus, and lateral tricuspid annulus locations. The atrial electromechanical connection (Pa’), defined as the time interval from the onset of the P wave on the surface electrocardiogram to the beginning of the late diastolic wave (a’), was captured from the lateral mitral (Pa’ lateral), septal mitral (Pa’ septal), and lateral tricuspid annuli (Pa’ tricuspid) (Figure 1a–c). The difference between Pa’ lateral and Pa’ tricuspid was termed as interatrial electromechanical delay, the difference between Pa’ lateral and Pa’ septal was defined as left intra-atrial electromechanical delay, and the difference between Pa’ septal and Pa’ tricuspid was defined as right intra-atrial electromechanical delay. The parameters were averaged over three sequential heartbeats.

### 2.4. Evaluation of Left Atrial Mechanical Functions

Simultaneously with the echocardiography, an electrocardiogram was recorded. Measurements were documented over three consecutive cardiac cycles. All atrial volumes were derived from the apical four-chamber perspective. When outlining the boundaries, we encompassed the left atrial walls, excluding the pulmonary veins and the left atrial appendage. The left atrial volume index (LAVI) was determined by dividing the left atrial volume by the body surface area of the patient. We computed the maximum left atrial volume (LAV_max_), the minimum left atrial volume (LAV_min_), and the precontraction left atrial volume (LAV_preA_) (Figure 2a–c).

The subsequent equations were utilized for calculating LA functions [8]:LA total emptying volume (LA_TEV_) = V_max_ − V_min_
LA ejection fraction (LA_EF_) = (V_max_ − V_min_/V_max_) × 100%
LA passive emptying volume (LA_PEV_) = V_max_ − V_preA_
LA active emptying volume (LA_AEV_) = V_preA_ − V_min_
LA active emptying fraction (LA_AEF_) = (V_preA_ − V_min_/V_preA_) × 100%

The LA ejection fraction (LA_EF_) characterizes the LA reservoir function, the LA passive emptying fraction describes the LA conduit function, and the LA active emptying fraction defines the LA pumping function.

### 2.5. Statistical Analysis

The SPSS (Statistical Package for Social Sciences) software version 20.0 for Windows (SPSS Inc., Chicago, IL, USA) was used to analyze the research data. Descriptive parameters were presented as mean ± standard deviation (range), frequency distribution, and percentage. Categorical parameters were assessed using the McNemar test. Parameters with a normal distribution were evaluated through both visual methods (histograms and probability plots) and analytical techniques (the Kolmogorov–Smirnov test). For variables that displayed an abnormal distribution, the Wilcoxon test was employed to establish statistically significant differences among parameters, while the Paired-Samples *t*-test was applied to normally distributed parameters. A *p* value of <0.05 was selected as the statistical significance level. According to the power analysis, the power of the study in the planned 68-patient study universe was evaluated as 90.09%.

## 3. Results

Our study included 68 female patients who were scheduled to receive chemotherapy for breast cancer or lymphoma. The mean age of the patients was 41.6 ± 7.9 years. None of the patients had a chronic disease history that required regular medication use other than their current illnesses. Additionally, none of our patients consumed alcohol or smoked. The basic demographic characteristics of the patients, the chemotherapeutic drugs administered, and the laboratory parameters are shown in Table 1.

In the routine echocardiographic assessment, there were noteworthy alterations in LV diastolic and systolic dimensions, septal and posterior wall thicknesses, LV mass index, LV ejection fraction, and LA anteroposterior diameter following the administration of chemotherapeutic agents, as contrasted with the pre-treatment measurements (Table 2). In the tissue Doppler echocardiographic evaluation, the mitral E wave, mitral lateral annulus “e” and “a” waves, the mitral mean “e” wave, and the tricuspid tissue Doppler “e” wave values decreased significantly compared to the values before administering chemotherapeutic drugs. There were no notable alterations noted in the remaining Doppler and tissue Doppler parameters (Table 3).

We did not observe atrial arrhythmia in any of the patients included in our study during the follow-up evaluation. Regarding the atrial electromechanical evaluation, there were significant variations between the pre- and post-treatment values for Pa’_lateral_, Pa’_septal_, and Pa’_tricuspid_, but there were no important variations in intra-left atrial EMD, intra-right atrial EMD, or interatrial EMD (Table 4).

When examining the phasic function of the left atrium post chemotherapy, there were notable differences observed in LAV_max_, LAV_min_, LAV_pre-A_, the LA reservoir (ejection) function, the LA active emptying fraction, the LA total emptying volume, and the LA passive emptying volume in comparison to their respective pre-treatment values. The left atrial volumes increased, the LA ejection and active emptying fractions decreased, and the LA total emptying and passive emptying volumes increased. However, there were no significant differences in other left atrial functions compared to the values before chemotherapeutic treatment (Table 5).

The intra-observer and interobserver correlation coefficients for echocardiographic parameters ranged from 0.62 to 0.93 (*p* < 0.05 for all).

## 4. Discussion

We observed atrial volume enlargement and electromechanical delay development following chemotherapy compared to pre-chemotherapy measures in our study, which included adult patients with no chronic health concerns other than their existing malignancies.

We did not observe atrial arrhythmia in any of the patients included in our study during the follow-up evaluation. We attributed this finding to the fact that the patient population we selected did not have any chronic diseases that could increase the risk of arrhythmia other than their existing malignancies, and that our patients were relatively young. This was an expected result since we planned the study design to minimize the risk of bias by keeping the exclusion criteria strict and to evaluate the pure effects of chemotherapeutic agents on left atrial functions as much as possible.

The number of patients diagnosed with cancer has significantly increased due to factors such as easier access to high-tech diagnostic techniques, widespread use of these techniques, and standardization of screening tests [9]. With the development of new-generation treatments, these patients can now live for many years, and some may even achieve complete remission for their current malignancies. However, as survival rates increase after treatment, the side effects of these chemotherapeutic agents, most of which are not specific for cancer cells, on almost all organs and systems in these patients have emerged as a new cause of morbidity and mortality [10]. The cardiovascular system is among the systems most affected by cardiotoxic chemotherapeutic agents, making cardiovascular diseases the most significant cause of morbidity and mortality in these patients during the post-treatment period [11]. The effects of cardiotoxic chemotherapeutic agents on ventricular function have been the subject of many scientific studies, yielding important findings and insights [12,13,14]. However, studies investigating the effects of these agents on the atria are limited. Despite being one of the most common arrhythmias, the scarcity and impracticality of methods that can be used for early diagnosis can still lead to delayed diagnosis and treatment of atrial fibrillation [3].

Atrial fibrillation occurs as a result of progressive atrial electromechanical impairment along with disorganized atrial activity. In many studies, an increase in atrial dimensions has been associated with an increase in the frequency of paroxysmal atrial fibrillation, which is attributed to the disorganized atrial electromechanical activity mentioned above [15,16]. In our study, statistically significant increases were observed in both the left atrial anteroposterior diameter and the left atrial volumes when compared with pre-chemotherapy findings.

One important aspect of atrial fibrillation is its persistence in an infinite loop after being triggered by any catalytic factor. The triggering catalytic factor is usually a combination of anatomical, cellular, mechanical, or electrical atrial remodeling mechanisms at certain levels [17,18,19]. The direct degenerative effects on the cell nucleus and cell membrane, along with the apoptosis and fibrosis-inducing mechanisms induced by cardiotoxic chemotherapeutic agents, are well known [12,18,20]. In a study involving 53 patients treated with cardiotoxic chemotherapeutic agents, a significant increase in the left ventricular end-systolic volume (LVESV) and a statistically significant decrease in the left ventricular ejection fraction (LVEF) were observed [21]. Similarly, in our study, we found statistically significant increases in LVEDD, LVESD, IVS, and PW values, along with a decrease in LVEF. Moreover, as mentioned above, we also observed significant increases in atrial dimensions and volumes, which inevitably indicate the outcome of the pathological mechanism that affects the entire cardiac structure at the cellular and molecular levels.

In a study comparing 34 patients with paroxysmal atrial fibrillation (PAF) to 31 healthy control subjects, significant increases in the left atrial size and volume were observed in PAF patients compared to the control group, along with intra-left atrial mechanical delay [15]. Additionally, in a retrospective study involving breast cancer patients treated with cardiotoxic chemotherapeutic agents, it was observed that the lateral PA duration increased in treated patients when compared to healthy volunteers [21]. In our study, we also found similar increases in lateral, septal, and tricuspid PA durations after chemotherapy; however, there was no statistically significant difference in terms of intra-atrial, interatrial, or intra-right atrial electromechanical delay.

In a study by Bayar et al. involving celiac disease patients, an increase in both the total emptying volume and the active emptying volume was observed [22]. In contrast, in our patients, we detected an increase in the left atrial total emptying volume and the passive emptying volume after chemotherapy, but we did not observe any significant change in the active emptying volume. Furthermore, the left atrial reservoir (ejection) function and pumping (active emptying fraction) function decreased after chemotherapy, while the left atrial conduit (passive emptying fraction) function remained unchanged compared to pre-treatment values. These findings were interpreted as indicating that post-chemotherapy patients who received cardiotoxic chemotherapeutic agents exhibited both left atrial dilatation and left atrial systolic dysfunction.

### Study Limitations

The primary limitation of our study was its small sample volume and relatively short follow-up period. In addition, since we aimed to evaluate the effect of the chemotherapy protocol on left atrial electromechanical functions as a whole, we did not evaluate the effect of each chemotherapeutic agent separately for all patients. This can be considered to be another limitation of our study.

## 5. Conclusions

A statistically significant impairment was detected in atrial electromechanical functions, which is an early indicator of atrial fibrillation and other atrial arrhythmias, in cancer patients treated with cardiotoxic chemotherapeutic agents. Measuring echocardiographic atrial electromechanical functions is an accessible, practical, and inexpensive way to evaluate the risk of future atrial arrhythmias. As one of the most common cardiac disorders and an important morbidity/mortality factor, early detection of atrial arrhythmias is very important for public health and the economy. However, in order for this early assessment method to be used frequently and even to be included in the guidelines, our findings need to be confirmed with much larger and longer-term follow-up studies and the increased frequency of long-term atrial arrhythmia in these patients needs to be clinically determined. We believe that future studies will meet these requirements.

## Figures and Tables

**Figure 1 medicina-60-01516-f001:**
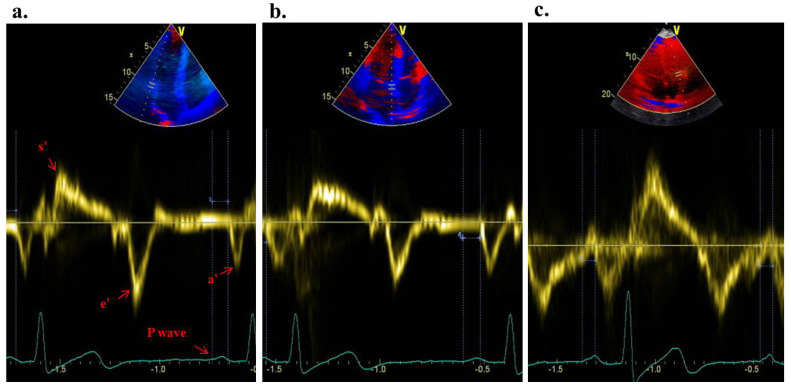
Atrial electromechanical coupling (Pa’), the time period from the beginning of the P-wave on the superficial electrocardiogram to the start of the late diastolic wave a’ on tissue Doppler echocardiography (mitral lateral annulus (**a**), septal annulus (**b**), and tricuspid lateral annulus (**c**)).

**Figure 2 medicina-60-01516-f002:**
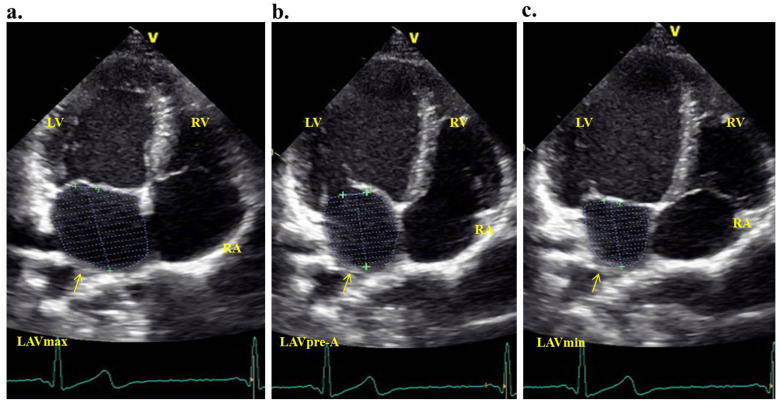
Maximum (**a**) atrial precontraction (**b**) and the minimum (**c**) volume measurements of the left atrium were made by transthoracic echocardiography on an apical four-chamber window (LV = left ventricule; RV = right ventricule; RA = right atrium; LAVmax = left atrium maximal volume; LAVpre-A = left atrial precontraction volume; LAVmin = left atrium minimum volume).

**Table 1 medicina-60-01516-t001:** Distribution of baseline clinic and laboratory characteristics in patients receiving cardiotoxic chemotherapeutic agents.

	Patients (*n* = 68) Mean ± SD Median (IQR) *
Age (year)	41.6±7.9
BMI (kg/m^2^)	25.9 ± 3.75
BSA (m^2^)	1.72 ± 0.15
Chemotherapeutic agents, n (%)	
Doxorubicin	67 (99)
Siklofosfomid	59 (87)
Paclitaxel	22 (32)
Transtuzumab	12 (18)
Docetaxel	6 (9)
Carboplatin	1 (1)
Adriamycin	3 (4)
5-fluorourasil	2 (3)
Hormone therapy	31 (46)
Radiotherapy	31 (46)
Glucose (mg/dL)	94.1 ± 12.7
BUN	11.2 (4.0)
Creatinine (mg/dL)	0.66 ± 0.10
Hemoglobin (g/dL)	12.7± 1.4
White blood cell (10^3^/mL)	7.4 ±2.2
Platelet (10^3^/mL)	289.9 ± 76.3
AST (Ul/L)	16.4 ± 5.1
ALT (Ul/L)	11.7 (7.1)

ALT = Alanine transaminase; AST = Aspartate transaminase; BMI = body mass index; BSA = body surface area; SD = standard deviation. * IQR = inter-quantile range.

**Table 2 medicina-60-01516-t002:** Distribution of blood pressure and basic echocardiographic parameters in patients receiving cardiotoxic chemotherapeutic agents.

	Before Cardiotoxic Chemotherapeutic Agents (n = 68) Mean ± SD Median (IQR) *	After Cardiotoxic Chemotherapeutic Agents (n = 68) Mean ± SD Median (IQR) *	*p* Value
Systolic BP (mmHg)	121 ± 11.2	125.6 ± 16.1	0.098
Diastolic BP (mmHg)	77.5 ± 8.4	79.3 ± 9.7	0.13
Heart Rate (bpm/min)	82.2 ± 10.8	84.4 ± 13.4	0.26
Ejection Fraction (%)	62 ± 3.8	60.2 ± 4.2	<0.001
LVEDD (mm)	43.1 ± 4.4	44.4 ± 4.3	<0.001
LVESD (mm)	25.3 ± 3.3	26.3 ± 3.5	<0.001
LVMI (g/m^2^)	82.3 ± 20.1	90.4 ± 21.8	<0.001
IVS (mm)	9.9 ± 1.4	10.1 ± 1.4	0.02
PW (mm)	9.7 ± 1.3	10.2 ± 1.4	<0.001
LA diameter (mm)	29.2 ± 3.5	30.2 ± 3.3	<0.001
RV (mm)	23.4 ± 4.0	23.2 ± 3.4	0.44

BP = blood pressure; IVS = inter-ventricular septum; LA = left atrium parasternal long axis; LVEDD = left ventricular end-diastolic diameter; LVESD = left ventricular end-systolic diameter; LVMI = left ventricular mass index; PW = posterior wall; RV = right ventricle; SD = standard deviation. * IQR = inter-quantile range

**Table 3 medicina-60-01516-t003:** Distribution of mitral and tricuspid leaflets and annular pulse-wave Doppler parameters in patients taking cardiotoxic chemotherapeutic drugs.

	Before Cardiotoxic Chemotherapeutic Agents (n = 68) Mean ± SD Median (IQR) *	After Cardiotoxic Chemotherapeutic Agents (n = 68) Mean ± SD Median (IQR) *	*p* Value
Mitral E (cm/s)	76.6 ± 17	71.0 ± 15.3	0.001
Mitral A (cm/s)	69.3 ± 13.7	68.4 ± 12.7	0.65
M. lateral annulus e’(cm/s)	15.4 ± 3.3	14.0 ± 3.0	0.001
M. lateral annulus a’(cm/s)	12.2 ± 2.6	11.1 ± 2.6	0.005
M. lateral annulus s’(cm/s)	9.9 ± 2.3	9.4 ± 2.6	0.16
M. septal annulus e’(cm/s)	11.5 ± 2.1	11.2 ± 2.3	0.29
M. septal annulus a’(cm/s)	10.3 ± 2.3	10.7 ± 2.5	0.21
M. septal annulus s’(cm/s)	8.9 ± 1.8	8.6 ± 2.0	0.29
Tricuspid annulus e’(cm/s)	15.3 ± 3.0	14.2 ± 2.9	0.003
Tricuspid annulus a’(cm/s)	16.5 ± 4.2	16.8 ± 4.1	0.67
Tricuspid annulus s’(cm/s)	13.6 ± 2.9	13.2 ± 2.1	0.32
Mitral E/A	1.2± 0.31	1.1 ± 0.32	0.09
Mitral E/e’	5.9 ± 1.2	5.8 ± 1.4	0.28
Mitral mean e’(cm/s)	13.5 ± 2.4	12.6 ± 1.2	0.006
Mitral mean a’(cm/s)	11.2 ± 2.1	10.9 ± 2.1	0.15
Mitral mean s’(cm/s)	9.4 ± 1.7	9.0 ± 1.9	0.09

* IQR = inter-quantile range.

**Table 4 medicina-60-01516-t004:** Atrial electrical activity parameters in patients taking cardiotoxic chemotherapeutic drugs.

	Before Cardiotoxic Chemotherapeutic Agents (n = 68) Mean ± SD Median (IQR) *	After Cardiotoxic Chemotherapeutic Agents (n = 68) Mean ± SD Median (IQR) *	*p* Value
Pa’ lateral (ms)	60.5± 11.6	68.8 ± 11.0	<0.001
Pa’ septal (ms)	55.1 ± 9.5	63.3 ± 10.0	<0.001
Pa’ tricuspid (ms)	58.3 ± 12.8	66.4 ± 11.7	<0.001
Interatrial-EMD (ms)	1 (16.75) *	1 (15.75) *	0.91
Intra-LA-EMD (ms)	5 (11.75) *	5 (14.75) *	0.95
Intra-RA-EMD (ms)	4 (12) *	4 (13) *	0.72

Pa’ = Time interval from the onset of the P-wave on the surface ECG to the peak of the late diastolic wave (a’); LA = Left atrium; RA = Right atrium; EMD = Electromechanical delay; SD = Standard deviation. * IQR = Inter quantile range

**Table 5 medicina-60-01516-t005:** Left atrial mechanical functions in patients taking cardiotoxic chemotherapeutic drugs.

	Before Cardiotoxic Chemotherapeutic Agents (n = 68) Mean ± SD Median (IQR) *	After Cardiotoxic Chemotherapeutic Agents (n = 68) Mean ± SD Median (IQR) *	*p* Value
LAVmax (mL/m^2^)	21.2 ± 5.8	25.2 ± 6.7	<0.001
LAVmin (mL/m^2^)	7.4 ± 3.4	9.3 ± 3.5	<0.001
LAVpre-A (mL/m^2^)	12.2 ± 4.0	14.2 ± 4.5	<0.001
LA reservoir (ejection) function %	65.5 ± 9.8	63.1 ± 9.9	0.03
LA conduit (passive emptying fraction) function %	42.3 ± 11.0	42.9 ± 11.6	0.65
LA pumping (active emptying fraction) function %	40.0 ± 12.85 40.1 (18)	33.7 ± 19.5 33.8 (19.3)	0.01 0.046
LA total emptying volume (mL/m^2^)	13.7 ± 3.8	15.9 ± 5.1	<0.001
LA passive emptying volume (mL/m^2^)	9.0 ± 3.4	11.0 ± 4.6	<0.001
LA active emptying volume (mL/m^2^)	4.75 ± 1.95	5.0 ± 2.96	0.54

* IQR = inter-quantile range; LA = left atrium; LAVmax = left atrium maximum volume; LAVmin = left atrium minimum volume; LAVpreA = left atrium volume before atrial systole; SD = standard deviation.

## Data Availability

The original contributions presented in this study are included in the article; further inquiries can be directed to the corresponding authors.

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
