# Peer review of "Evaluation of Left Atrial Electromechanical Delay and Left Atrial Phasic Functions in Patients Undergoing Treatment with Cardiotoxic Chemotherapeutic Agents"

_medicina, 2024, doi:10.3390/medicina60091516_

Round 1

Reviewer 1 Report

Comments and Suggestions for Authors

This study aims to determine how cardiotoxic chemotherapeutic drugs affect the mechanical and electrical properties of the left atrium in patients and estimate the risk of atrial arrhythmia after treatment. It makes a useful link between echocardiographic tests and possible clinical effects in cancer and cardiology.

Introduction and Objectives: 

• Strengths: The opening shows how important it is to understand atrial participation in patients being treated with cardiotoxic chemotherapy drugs, and the review of relevant literature about ventricular measurements backs this up.

• Weaknesses: The exact study gap could have been made more evident. The opening talks about how few studies have been done on atrial effects, but it would be better if it went into more detail about what this study adds to what has already been learned.

Methods

• Strengths: The study procedure is well-written and includes clear criteria for who can and cannot participate, thorough explanations of the echocardiographic methods, and a well-documented process for getting ethical approval.

• Weaknesses: The methods are usually robust, but more details on the statistical power and whether the sample size was big enough to find significant differences would help people understand how vital the study's statistics are.

Results

• Strengths: The detailed results fully examine how cardiac measures changed before and after chemotherapy. It is easier to understand the data when it is shown in clear tables and detailed figures.

• Weaknesses: The results part could use more in-depth statistical analysis. For example, it could use changes for multiple comparisons or confidence intervals to help readers understand how different the results are and how important they are.

Discussion

• Strengths: The talk makes a good link between the study's results and possible therapeutic consequences, especially the chance of developing atrial rhythms. It also connects the results to bigger-picture physiological and neurological situations.

Problems: The conversation could discuss other possible reasons for the changes that were seen, such as things unique to the patient or other treatments they received. More detail about the study's limits, such as possible selection biases or the fact that it was based on observations, would also give a more fair view.

Conclusion

• Strengths: The end does a good job of summarizing the results and their meaning, highlighting how important it is to monitor cardiac functions in chemotherapy patients.

• Weaknesses: It could be better if it gave specific advice for professional practice or ideas for future research, like ongoing studies to see how these changes happen over a longer time or in a bigger group of people.

Reviewer 2 Report

Comments and Suggestions for Authors

This study is an interesting investigation into the chemotherapy-induced cardiomyopathy, particularly its effects on left atrial function, from the perspective of echocardiography. The content is straightforward and very clear, but there are several points that need to be addressed for improvement:

1)    The study mentions the incidence of atrial fibrillation (Af) as an objective. However, it is necessary to discuss deeper into what specific issues the comorbidity of Af in chemotherapy-induced cardiomyopathy.

2)    Why was a period of one month chosen for the pre- and post-chemotherapy evaluation?

3)    Did the incidence of Af actually increase? This should be assessed. As part of this discussion, evaluating left atrial remodeling through echocardiography would seem to be a natural flow.

4)    The details of the chemotherapy administered have not been described. Were the dosage and regimen exactly the same for all patients?

5)    Did you evaluate the relationship between the type of chemotherapy and left atrial remodeling?

6)    There are reports that heart rate increases in cases of chemotherapy-induced cardiomyopathy. How did blood pressure and heart rate change in your study?"

Reviewer 3 Report

Comments and Suggestions for Authors

This is an original manuscript with a nice observation. Certain chemotherapeutic medications have the potential to cause or worsen atrial arrhythmias when used in the treatment of cancer. These medications may produce electrolyte imbalances, interfere with the heart's autonomic nervous system, or directly damage the heart muscle (cardiotoxicity), all of which increase the risk of arrhythmias.

Tyrosine kinase inhibitors, anthracyclines (such as doxorubicin), and other targeted medicines used in the treatment of cancer, for instance, have been linked to an increased risk of arrhythmias, including atrial arrhythmias. Healthcare professionals treating cancer patients on these medications must continuously monitor their heart function and treat any arrhythmias that develop. Authors are requested to check the signature of electromechanical properties by the use of anticoagulant drug prior to chemotherapy. 

I hope that's not possible immediately, but if it is, the author should consider for future. 

Round 2

Reviewer 2 Report

Comments and Suggestions for Authors

While some questions were addressed, others were not. We would have appreciated individual responses to each comment.